# Barriers to cleaning of shared latrines in slums of Addis Ababa, Ethiopia

**Kidist Hailu**[1]☯, **Zewdie Aderaw Alemu**[2,3], **Metadel Adane**[4]☯ *

1 Infection Prevention and Control (IPC) Unit, Bethzatha General Hospital, Addis Ababa, Ethiopia,
2 Department of Public Health, Health Sciences College, Debre Markos University, Debre Markos, Ethiopia,
3 GAMBY Medical and Business College, Addis Ababa, Ethiopia, 4 Department of Environmental Health,
College of Medicine and Health Sciences, Wollo University, Dessie, Ethiopia

☯ These authors contributed equally to this work.
* metadel.adane2@gmail.com

**Data Availability Statement:** All relevant data are in the paper and its Supporting Information files.

**Funding:** The author(s) received no specific funding for this work.

## Abstract

Shared latrines and other shared sanitation facilities are vital for communities that lack private latrines. However, the cleanliness of shared latrines continues to be a problem in sub-Saharan Africa, including slums of Addis Ababa, Ethiopia. Investigating the barriers to cleaning of shared latrines may inform the future strengthening of comprehensive sanitation programs in slums of Addis Ababa, Ethiopia. Thus, a community-based unmatched case–control study was conducted among 100 case and 200 control households that were users of shared latrines from September to November 2017 in a slum district in Addis Ababa. Cases were those who had not cleaned their shared latrines and controls were those who had cleaned their shared latrines at least once during the week prior to data collection. Data were collected using a structured questionnaire and an on-the-spot-observational checklist and analyzed using bivariate (crude odds ratio [COD]) and multivariable (adjusted odds ratio [AOR]) unconditional logistic regression model. Variables having a $p$-value of less than 0.25 from the bivariate logistic regression analysis were retained into multivariable analysis. From the multivariable analysis, variables with $p<0.05$ were declared as factors significantly associated with barriers to cleaning of shared latrines. We found that about half 99 (49.5%) of shared latrines used by cases and almost one-third 32 (32.0%) of the shared latrines used by controls had visible cracks and spaces in the floor and slabs. The barriers to cleaning of shared latrines were found to be monthly household income of less than $55.60 USD (AOR = 1.80; 95%CI: 1.2–3.10), users feeling a lack of privacy during latrine use (AOR = 2.95; 95% CI: 1.60–5.43), no locking latch on the latrine door (AOR = 4.60; 95% CI: 2.43–8.79), inadequate ventilation of latrine (AOR: 4.88; 95% CI: 2.44–9.63), lack of regular monitoring of latrine by health extension workers (AOR = 2.86; 95%CI: 1.32–6.21) and a lack of enough water at home for cleaning the latrine (AOR = 4.91; 95% CI: 1.07–9.48). This study found several barriers to cleaning of shared latrines in slums of Addis Ababa. We recommend that stakeholders promote cleaning of shared latrines by designing programs to improve latrine privacy by adding or modifying the superstructure and including a door with locking latch, to make adjustments to the structure for better ventilation, to ensure regular monitoring of latrines by health extension workers and to make enough water consistently available for regular latrine cleaning.

**Competing interests:** The authors have declared that no competing interests exist.

**Abbreviations:** AOR, adjusted odds ratio; COR, crude odds ratio; CI, confidence interval; VIF, variance inflation factor.

# Background

Rapid expansion of urbanization in developing countries has resulted the creation of urban slums [1, 2], which are characterized by one or more of the features of overcrowding, inadequate safe water supply, insecure property tenure, inadequate drainage and sewage networks, and lack of sanitation or proper solid waste disposal [2–6].

Shared latrines, including public latrines, are common in African slums, while a lack of space for construction of private latrines is a big challenge [1, 5, 7]. During the rainy season experienced by many African countries when unsealed shared latrines in slums may overflow and disperse pathogens [8] and diarrhea incidence in slums tends to peak [9], higher diarrhea transmission occurs among children under five than during the dry seasons. Furthermore, lack of cleaning of shared latrines in slums worsens the already bad sanitation situation [10].

Heijenen *et al.* reported that shared latrines were generally in poorer condition than those latrines that were not shared [11, 12]. A shared latrine can be equated with a common good whose management depends on the users. If users do not work together towards keeping the facilities clean, the quality and utilization of the latrine may decline [13].

In Africa's urban slums where private latrines are lacking, there is a great demand for public latrines. Therefore, improving their hygienic status is vital for minimizing disease outbreaks in these areas [3]. Bartlett (2003) found that the lack of latrines in poor communities caused many people to defecate in the open or into plastic bags and papers that were then discarded with the household garbage [14]. A recent study in eastern Ethiopia found a lack of effective social mobilization to be the main cause for open defecation [15].

Addis Ababa is an Ethiopian example of a city characterized by overcrowded settlements with shared latrines that are in poor hygienic condition [16, 17]. This situation increases the risk of diarrheal and other diseases [18, 19]. For example, in 2008, the city-wide data on basic indicators in Addis Ababa reported that 26.0% of the houses and the majority of slum dwellers had no toilet facility, 33.0% of households shared a latrine among more than six households and 71.0% of the households did not have adequate sanitation [16]. Furthermore, studies in slums of Addis Ababa indicated that poorly maintained shared latrines may have contributed to contamination of household water supplies as a result of the proximity of many latrines to crowded neighborhoods [19, 20].

The WHO/UNICEF Joint Monitoring Program (JMP) excludes what it calls 'shared' sanitation facilities—those used by two or more households—from the definition of 'improved sanitation' due to the concern that shared toilets tend to be less hygienic than private ones [21]. However, there is limited existing information on why shared latrines are not cleaned by users in slum contexts. A previous study done in slums of Addis Ababa found that sanitation and hygienic levels of shared latrines were low, although the main reasons for this could not be addressed [19]. Therefore, the purpose of this study was to address this gap in knowledge by identifying factors significantly associated with barriers to cleaning of shared latrines in slums of Addis Ababa, Ethiopia. To that end, this unmatched case–control study was conducted using data collected by on-the-spot-observation and interviews of study participants who reported being primarily responsible for cleaning of the shared latrines.

# Method

## Description of the study area

The study was conducted in District 05 in Lideta Sub-City, Addis Ababa. Addis Ababa has 116 districts (the lowest administrative unit in Addis Ababa). In the absence of a clear demarcation of the slums in Addis Ababa, the UN-HABITAT study in 2010 estimated that four-fifths (80%)

of Addis Ababa was a slum, where most residents lived in houses rented from the local government [17]. A previous study in slums of Addis Ababa reported that 51.0% of slum residents used public latrines and 26.8% of them also used pit latrines that were without a slab floor [19]. This study was conducted among shared latrine users, which included users of public latrines. Shared latrines were those that were shared by two or more households.

## Study design and study period

A community-based unmatched case–control study was conducted from September to November, 2017. Cases were those who had not cleaned a shared latrine and controls were those who had cleaned a shared latrine at least once during the week prior to data collection based on self-report. Case participants and control participants did not share the same latrines.

## Source populations, inclusion and exclusion criteria

The source population was all dwellers in District 05 in Lideta Sub-City, Addis Ababa who had used shared latrines during the week prior to data collection, whereas the study population was selected cases and controls among District 05 residents in Lideta Sub-City who had used shared latrines during the week prior to data collection. In this study, latrine users who were responsible for the cleaning of their latrines were included, whereas latrine users who were not responsible for the cleaning of their latrines were excluded. Households that had private latrines and did not share with others were excluded.

## Sample size determination

The sample size was determined using Epi Info version 7.0 with the consideration of proportion ($p$) of households that had not cleaned a shared latrine in the one week prior to data collection taken as 50.0% due to lack of studies in a similar setting, and expected odds ratio of 1.5 used based on another study suggestion to obtain adequate sample size [22]. Furthermore, power of 80.0% and control-to-case ratio of 2:1 was also used in line with the Pitman Efficiency assumption [23]. Considering a 10% non-response rate, the final adequate sample size becomes 100 cases and 200 controls, for a total of 300 households of users of shared latrines.

## Case and control study participant recruitment

By house-to-house visits, cases were selected from those who had not cleaned the shared latrine at least once during the week prior to data collection and controls were selected from shared latrine users who had cleaned the latrine at least once during the week prior to data collection. The selection of study participant cases and controls was performed randomly after the participants self-reported that either they had cleaned (controls) or not cleaned (cases) the shared latrine.

Study participants who were not available during the survey were revisited once on the same day or the next day. If not available again, a third visit was made. If not available the third time, the study participant would have been considered a non-respondent. However, in this study there was no non-respondent, which might be due to the random selection of cases and controls based on the self-report of latrine cleaning condition (cleaned or not cleaned), before the sample size was achieved. Since most of the slum dwellers were day laborers and the study area was business areas, the data was collected every weekend on Saturdays at mid-day and Sunday afternoons, times when individuals were more likely to be at home and available to participate in the survey.

## Operational definitions

**Slums**: Areas dominated by informal settlement that are characterized by one or more of the five characteristics of overcrowding, poor sanitation, insecure land tenure, lack of access to water supply, poor housing quality and other infrastructure [2].

**Shared latrines**: Refers to latrines shared by two or more households, including public latrines that are used by known households with a shared responsibility for cleaning by the users, whereas public latrines publicly accessible for everyone at a market, school or main road were excluded from this study.

**Latrine access**: Means that individuals accessed only the latrines that were within a certain distance from their homes as provided by *kebeles* for shared use, or use of their own shared latrine, or use of a latrine that was part of a household and also shared by two or more households.

**Barriers to cleaning of shared latrines**: Obstacles that prevented users from cleaning shared latrines.

**Cases**: Users of shared latrines who self-reported that they had not cleaned the latrine during the week prior to data collection.

**Controls**: Users of shared latrines who self-reported that they had cleaned the latrine at least once during the week prior to data collection.

## Data collection by observation and interview

Household survey data were collected using a pre-tested questionnaire and an observational checklist. The questionnaire and the observational checklist were adapted from various published literature [3, 24–28] and WHO and UN-HABITAT reports [1, 16, 29, 30]. The survey tool was first prepared in English (SI) and then translated into (local language) Amharic (SII) for use by participating households. Three data collectors who were BSc professionals in environmental health administered the survey. Data collectors were trained by the principal investigator for two days on the aim of the study, the content of the questionnaire and observation checklist, ethical issues and approaches during data collection.

Variables measured by interview included age, sex, marital status, educational status, occupation, monthly household income, household size, number of rooms in the house, number of households sharing latrine, privacy status during use of shared latrine, presence or absence of users participating collectively in decision making, presence or absence of monitoring of the latrines by health extension workers, latrine considered to be clean or not by users and availability of water at home for cleaning the latrine.

To collect data by direct observation, study participants were asked to show their latrine to the data collector. Data measured by on-the-spot-observation included superstructure materials, presence or absence of latrine door, locking latch and slab, status of latrine pit fullness, condition of slab (cracked and/or broken), ventilation condition of latrine, availability of handwashing facilities inside and/or near the latrine, presence of water in the handwashing facilities, availability of soap near the handwashing facilities.

Daily supervision was carried out by one supervisor and the principal investigator to check the completeness of the questionnaires and consistency of the data. When there was any missing data, correction was made by re-visiting the participant during the same day or the next day.

## Data quality assurance

To ensure the quality of the data, we also pre-tested the questionnaire among 10 cases and 20 control households (10% of the sample size) in one non-selected area of District 4 in Lideta Sub-City to evaluate its face and content validity. During the pre-test, face validity was verified by checking that the questions measured what they were intended to measure. Content validity

was also checked by three environmental health professionals who evaluated whether the survey contained questions that covered all aspects of the contents being measured. Data collection began after we approved the face and content validity of the survey tool.

Any amendment made to the questionnaire such as elimination of unnecessary questions, revision of confusing terms, and addition of important questions was based on the pre-test. Inter-observer reliability was ensured by providing clear definitions during training about shared latrines, cases and controls, latrine cleaning, superstructure of the latrine and events to be recorded and by providing feedback about discrepancies during daily supervision.

Also, 10% of the study participants were re-interviewed by another interviewer to check reliability of the information collected by different interviewers. The qualifications of the interviewers and the training they received also reduced the likelihood that interviewer bias occurred. Entered data was re-entered on 10.0% of the sample to check consistency and reduce errors of data entry. After the data was entered and cleaned, quality assurance measures were employed using descriptive statistics from cross-tabulation and frequency distribution before performing statistical analysis.

## Data management and analysis

The collected data were coded and entered in to EpiData version 3.1 and exported to Statistical Package for the Social Sciences (SPSS) version 23.0 for cleaning and analysis. Descriptive statistics were carried out, including means ±SD (standard deviations) for continuous variables. The presence of multi-collinearity among independent variables was checked using variance inflation factor (VIF). We found a maximum VIF of 2.0, which indicated there was no multicollinearity between independent variables.

Unconditional logistic regression model was used for data analysis. The modeling strategy involved estimating the bivariate analysis (crude odds ratio [COR]) and multivariable analysis (adjusted odds ratio [AOR]) at 95% CI. For selection to the final model, variables with a $p$-value less than 0.25 from bivariate analysis were included to the multivariable analysis. From the adjusted analysis, variables with $p < 0.05$ were taken as statistically significant and independently associated with barriers to cleaning of shared latrines in slums of Addis Ababa. The goodness-of-fit of the model was checked using the Hosmer-Lemeshow statistic [31], finding a $p$-value of 0.937, indicated the model was fit.

## Ethical considerations

Ethical clearance was obtained from the Institutional Review Board (IRB) of GAMBY Medical and Business College, Addis Ababa, Ethiopia. Permission to undertake this study was also obtained from Addis Ababa Health Bureau, slum District 05 in Lideta Sub-City. Written consent was obtained during recruitment of the study participants. During data collection, study participants who were using unclean latrines were advised to keep their latrine in a hygienic condition. Study participants also were informed that participation was affirmed by the procedure of probability sampling technique which provides an equal chance of selection. Confidentiality of the study was maintained by establishing codes instead of using names.

## Results

### Socio-demographic and economic characteristics of case and control households

This study included 300 study participants (100 cases and 200 controls) and had a response rate of 100%. Of the study participants, 49 (49.0%) of cases and 88 (44.0%) of controls were

**Table 1. Socio-demographic and economic characteristics among case and control study participants in slums of District 05, Lideta Sub-City, Addis Ababa, Ethiopia.**

| Variable | Category | Case (*N* = 100) | Control (*N* = 200) | COR (95% CI) |
|---|---|---|---|---|
| | | *n*(%) | *n*(%) | |
| Age (years) | 25–34 | 23(23.0) | 52(26.0) | 0.7(0.4–1.4) |
| | 35–44 | 34(34.0) | 73(36.5) | 0.8(0.5–1.4) |
| | >44 | 43(43.0) | 75(37.5) | Ref |
| Sex | Male | 49(49.0) | 88(44.0) | 0.8(0.5–1.3) |
| | Female | 51(51.0) | 112(56.0) | Ref |
| Marital status | Married | 76(76.0) | 152(76.0) | 1.6(0.8–3.5) |
| | Single | 15(15.0) | 18(9.0) | 0.5(0.2–1.3) |
| | Widowed | 5(5.0) | 21(10.5) | 0.9(0.3–3.0) |
| | Divorced | 4(4.0) | 9(4.5) | Ref |
| Educational status | Illiterate | 48(48.0) | 130(65.0) | 0.3(0.1–1.1) |
| | Read and write | 22(22.0) | 47(23.5) | 0.4(0.1–1.4) |
| | Elementary | 24(24.0) | 18(9.0) | 1.1(0.3–4.2) |
| | Secondary or above | 6(6.0) | 5(2.5) | Ref |
| Occupation | Housewife | 44(44.0) | 90(45.0) | 0.9(0.4–2.0) |
| | Government employee | 33(33.0) | 59(29.5) | 1.1(0.4–2.3) |
| | Daily laborer | 12(12.0) | 31(15.5) | 0.7(0.3–1.9) |
| | Merchant | 11(11.0) | 20(10.0) | Ref |
| Monthly household income ($US) | Less than $55.60 US | 27(27.0) | 89(44.5) | 2.2(1.3–3.6) |
| | $55.60 US or more | 73(73.0) | 111(55.5) | Ref |
| Household size (persons) | 6 or more persons | 24(24.0) | 43(21.5) | 1.1(0.6–2.0) |
| | 1–5 persons | 76(76.0) | 157(78.5) | Ref |
| House ownership | Rented from government (*kebele*) | 63(63.0) | 133(66.5) | 1.1(0.6–0.2) |
| | Privately rented | 21(21.0) | 40(20.0) | 1.2(0.6–2.5) |
| | Owned by householder | 16(16.0) | 27(13.5) | Ref |
| Number of rooms in house | ≤2 | 60(60.0) | 144(72.0) | 1.7(1.1–2.8) |
| | >2 | 40(40.0) | 56(28.0) | Ref |

1, Reference category; COR, Crude odds ratio

*The average exchange rate for $1 USD was 20.0 birr from September to November 2017.

males. The educational level of almost half 48 (48.0%) of cases and two-thirds 130 (65.0%) of controls were illiterate. The average monthly household income was $55.60 USD (United States Dollars); and 27 (27.0%) of cases and 89 (44.5%) of controls had less than $55.60 USD monthly household income. Of all households, about (two-thirds, 133 (66.0%) controls and 63 (63.0%) cases, rented government houses administered by *kebeles* (Table 1).

## The superstructure and privacy-related characteristics of the shared latrines among cases and controls

About two-thirds 62 (62.0%) cases and 143 (71.5%) controls shared one latrine among 11 to 13 households, whereas 23 (23.0%) cases and 37 (18.5%) controls shared one latrine among 6 to 10 households. About one-third 30 (30.0%) of case household latrines and one-fifth 43 (21.5%) of control household latrines had a superstructure constructed with bricks or stones. About one-tenth 13 (13.0%) of the case households and more than half 107 (53.7%) of controls shared latrines that had no door. Two-thirds 67 (67.0%) of cases and three-fourths 157 (78.5%) of controls mentioned lack of privacy when using the latrine was a problem (Table 2).

**Table 2. Superstructure and privacy-related characteristics of shared latrines among case and control study participants in slums of District 05, Lideta Sub-City, Addis Ababa, Ethiopia.**

| Characteristics | Category | Case (*N* = 100) | Control (*N* = 200) | COR (95%CI) |
|---|---|---|---|---|
| | | *n*(%) | *n*(%) | |
| Number of households sharing latrine | ≤5 | 15(15.0) | 20(10.0) | Ref |
| | 6–10 | 23(23.0) | 37(18.5) | 0.8(0.4–1.9) |
| | 11–13 | 62(62.0) | 143(71.5) | 0.6(0.3–1.2) |
| Superstructure materials of shared latrine | Bricks/stone | 30(30.0) | 43(21.5) | 1.1(0.60–2.1) |
| | Mud/wood | 5(5.0) | 41(20.5) | 0.9(0.15–2.0) |
| | Corrugated iron sheets | 65(65.0) | 121(58.0) | Ref |
| Adequate privacy during use of shared latrine | No | 33(33.0) | 157(78.5) | 7.4(4.3–12.6) |
| | Yes | 67(67.0) | 43(21.5) | Ref |
| Shared latrine had a door | No | 13(13.0) | 107(53.7) | 7.7(4.0–14.7) |
| | Yes | 87(87.0) | 93(46.5) | Ref |
| Shared latrine had door with a locking latch | No | 23(23.0) | 136(68.0) | 7.1(4.1–12.3) |
| | Yes | 77(77.0) | 64(32.0) | Ref |
| Shared latrine had good ventilation | No | 21(21.0) | 143(71.5) | 9.4(5.3–16.7) |
| | Yes | 79(79.0) | 57(28.5) | Ref |

## Shared latrine slab characteristics, latrine monitoring and collective decision-making practices

About 32 (32.0%) of the case households' latrines and nearly half of control households' 99 (49.5%) latrines had cracked or broken slabs. Almost two-thirds 121 (60.5%) of control households and half 53 (53.0%) of case households reported conflicts among latrine users. One-quarter 26 (26.0%) of case and four-fifths 166 (83.0%) of control households' latrines were not monitored regularly by health extension workers. Most control households 174 (87.0%), but only 31 (31.0%) of the case households, considered their latrines not clean (Table 3).

**Table 3. Latrine slab, monitoring practices and other characteristics of shared latrines among case and control study participants in slums of District 05, Lideta Sub-City, Addis Ababa, Ethiopia.**

| Characteristics | Category | Case (*N* = 100) | Control (*N* = 200) | COR (95%CI) |
|---|---|---|---|---|
| | | n(%) | n(%) | |
| Cracked or broken slab | No | 68(68.0) | 101(50.5) | 2.1(1.3–3.4) |
| | Yes | 32(32.0) | 99(49.5) | Ref |
| The latrine pit was full | No | 75(75.0) | 129(64.5) | 0.6(0.3–1.0) |
| | Yes | 25(25.0) | 71(35.5) | Ref |
| Users participated collectively in decision making | No | 23(23.0) | 147(73.5) | 9.3(5.3–16.3) |
| | Yes | 77(77.0) | 53(26.5) | Ref |
| Users experienced conflict | Yes | 53(53.0) | 121(60.5) | 1.4(0.8–2.2) |
| | No | 47(47.0) | 79(39.5) | Ref |
| Regular monitoring of the latrines by health extension workers | No | 26(26.0) | 166(83.0) | 13.9(7.8–24.8) |
| | Yes | 74(74.0) | 34(17.0) | Ref |
| Latrine was considered to be clean by users | Yes | 69(69.0) | 26(13.0) | 14.9(8.2–26.9) |
| | No | 31(31.0) | 174(87.0) | Ref |

Ref, Reference category.

**Table 4. Availability of water and handwashing facilities in shared latrines among case and control study participants in slums of District 05 in Lideta Sub-City, Addis Ababa, Ethiopia.**

| Variables | Category | Case (N = 100) | Control (N = 200) | COR (95%CI) |
|---|---|---|---|---|
| | | n(%) | n(%) | |
| Enough water was available at home for cleaning the latrine¥ | No | 69(69.0) | 194(97.0) | 0.7(0.8–6.3) |
| | Yes | 31(31.0) | 6(3.0) | Ref |
| Handwashing facilities inside and/or near the latrine* | No | 84(84.0) | 165(82.5) | 1.1(0.7–2.5) |
| | Yes | 16(16.0) | 35(17.5) | Ref |
| Presence of water in the handwashing facilities* | No | 12(75.0) | 27(77.1) | 0.8(0.9–1.7) |
| | Yes | 4(25.0) | 8(22.9) | Ref |
| Availability of soap near the handwashing facilities* | No | 14(87.5) | 30(85.7) | 1.16(0.8–1.2) |
| | Yes | 2(12.5) | 5(14.3) | Ref |

Ref, Reference category.

*Not included during the logistic regression analysis due to the presence of zero frequency either in the case or control households.

¥The quantity of the available water was not measured and only the perception that they have enough water at home for latrine cleaning was studied.

## Availability of water and handwashing facilities in the shared latrines

Nearly one-third 31 (31.0%) of control household had enough water available within or near the latrine for cleaning the sanitation facilities. A large majority of case 84 (84.0%) and control 165 (82.5%) latrines had no handwashing facilities inside and/or near the latrine. Of those case households' latrines that had handwashing facilities inside and/or near the latrine 16 (16.0%), only 4 (25.0%) had water and 2 (12.5%) had water and soap (Table 4).

## Barriers associated with not cleaning shared latrines

This study identified six barriers significantly associated with not cleaning shared latrines in slums of District 05 in Lideta Sub-City, Addis Ababa. Householders that had a monthly income of less than $55.60 USD were 1.80 times (AOR = 1.80; 95% CI: 1.2–3.10) higher not to cleaned the shared latrine the previous week than those that had a monthly income of $55.60 USD or above. The odds developing not to cleaned latrines among shared latrine users who felt a lack of privacy when using the latrines were 2.95 times (AOR = 2.95; 95% CI: 1.60–5.43) higher than those who had a feeling of privacy during use. The odds of developing not to cleaned the shared latrines among users of shared latrines without a locking latch for the latrines were 4.6 times (AOR = 4.60; 95% CI: 2.43–8.79) higher compared to those users of a shared latrine that had a locking latch.

The odds of developing not to cleaned shared latrines among householders using shared latrines that were without adequate ventilation were 4.88 times (AOR: 4.88; 95% CI: 2.44–9.63) higher than those who used a shared latrine that had adequate ventilation. The odds of developing not to cleaned shared latrines among users of shared latrines not monitored regularly by health extension workers were 2.86 times (AOR = 2.86; 95% CI: 1.32–6.21) higher than those using shared latrines that were monitored by health extension workers. The odds of developing not to cleaned shared latrines among users who believed that there was not enough water at home to clean the latrine were 4.91 times (AOR = 4.91; 95% CI: 1.07–9.48) higher than those who believed there was enough water at home to clean the latrine (Table 5).

## Discussion

The main aim of this study was to determine the barriers to cleaning of shared latrines in slums of District 05, Lideta Sub-City of Addis Ababa, Ethiopia. This unmatched case–control

**Table 5. Barriers associated with cleaning of shared latrines from multivariable logistic regression analysis in slums of District 05, Lideta Sub-City, Addis Ababa, Ethiopia.**

| Variables | Category | Cases (N = 100) | Controls (N = 200) | AOR (95%CI) |
|---|---|---|---|---|
| | | n(%) | n(%) | |
| Household monthly income (USD) | Less than $55.60 | 27(27.0) | 89(44.5) | 1.80(1.20–3.10) |
| | $55.60 or more | 73(73.0) | 111(55.5) | Ref |
| Feeling of privacy when using latrine | No | 33(33.0) | 157(78.5) | 2.95(1.60–5.43) |
| | Yes | 67(67.0) | 43(21.5) | Ref |
| Latrine door had a locking latch | No | 23(23.0) | 136(68.0) | 4.60(2.43–8.79) |
| | Yes | 77(77.0) | 64(32.0) | Ref |
| Latrine had adequate ventilation | No | 21(21.0) | 143(71.5) | 4.88(2.44–9.63) |
| | Yes | 79(79.0) | 57(28.5) | Ref |
| Regular monitoring of the latrine by health extension workers[¥] | No | 26(26.0) | 166(83.0) | 2.86(1.32–6.21) |
| | Yes | 74(74.0) | 34(17.0) | Ref |
| Enough water was available at home for cleaning the latrine[*] | No | 69(69.0) | 194(97.0) | 4.91(1.07–9.48) |
| | Yes | 31(31.0) | 6(3.0) | Ref |

Ref, Reference category.

[¥] Monitoring by health extension workers (HEWs) were done by regular visit to the latrine and feedback provided to users about keeping the latrine clean.

study found that the barriers to cleaning of shared latrines were monthly household income, users feeling a lack of privacy during latrine use, a shared latrine without locking latch, shared latrine without adequate ventilation, shared latrine not regularly monitored by health extension workers and a lack of enough water at home for cleaning the latrine.

Our findings showed an average household monthly income of less than $55.60 USD was inversely related to cleaning of shared latrines. This might be due to slum residents, who are commonly poor, giving attention to their food security concern rather than the hygienic conditions of shared latrines. A previous study in slums of Addis Ababa found nearly 14.1% of the wealthiest households had access to an improved latrine (not shared) compared to only 0.5% of the poorest who had their own unshared latrine [18]. Furthermore, due to a lack of income, the affordability of latrine cleaning agents might also be a challenge for slum dwellers. Low income status may hinder the purchase of latrine-cleaning and handwashing agents [32].

A lack of privacy was one of the barriers to cleaning of the shared latrines, which might be attributed to the structure being built of poor materials that provided less privacy. A study in Nepal reported that a lack of privacy when using the latrine pushed women to look for open defecation places [33]. Therefore, privacy is an important factor for both latrine use and cleaning. Lack of privacy might be due to the poor cleanliness of the latrine. A study in Kisumu, Kenya that found that latrines constructed with iron sheets, which were likely to have a wooden slab, tended to be more dirty than facilities built of bricks and having a cement floor [28]. The shared latrine without locking latch was also associated with a lack of latrine cleaning in our study, in contrast with the finding of a study in Tanzania that revealed non-shared latrines were less likely to have lockable doors than shared latrines [34]. The presence of a locking latch may increase the sense of ownership among those sharing the latrine. If the shared latrine users have their own keys, others will not use it.

A study in the Kibera slums in Nairobi found that respondents used shared latrines due to the scarcity of private household latrines and the poor condition of other sanitation facilities in that crowded area [35]. Consistent with our findings, Heijenen *et al.* reported that households sharing sanitation facilities were generally poorer than those that did not share, not necessarily because of sharing sanitation facilities but because of poverty [36]. A study in slums of Uganda

showed that 68% of the shared latrines were observed to be significantly dirty [25]. Another study has revealed that shared sanitation facility users are not committed to cleaning shared latrines [26]. In our study, a latrine door having no locking latch was one of the barriers to latrine cleaning, since that allowed it be used by people other than the specified users; this, in turn, may compromise the hygiene of the latrine.

Disposal of garbage close to homes was a significant risk factor for high fly densities and the presence of flies around the shared latrines [37]. A similar study in Rajshahi City slums in Bangladesh revealed that feces were observed around 61.0% of the latrines [38]. Unsanitary condition of latrines and poor hygiene behavior were related to increased diarrhea episodes in slums of An-Nasr in Jordan, Tebbaneh in Libya [39] and Ikare-Akoko in Nigeria [40]. Findings that substandard latrine construction contributed to the presence of flies indicate that improved superstructures may decrease fly densities around latrines [41, 42].

Our findings showed that a lack of regular monitoring of the latrine by health extension workers was a barrier to latrine cleaning, which might be due to the fact that when there is monitoring, users are concerned about implementing the country's health extension programs, which includes water, sanitation and hygiene programs. Other studies have also showed that the presence of a health extension program in Ethiopia changed the performance of health systems [43, 44]. The findings of our study and previous studies in Ethiopia have consistently shown that the benefit of a health extension program is immense and that consistent follow-up by health extension workers is associated with improvement of the hygienic and sanitation condition of latrines. The frequency of such latrine visits depends on the observed cleanliness of the latrine. For households that maintain clean latrines, the frequency of visits is low, whereas repeated visits can be made to households that do not clean the latrine regularly.

This study also indicates that a lack of water availability is one of the barriers to cleaning shared latrines. In our study setting, the source of the water for all households was the government. However, due to the presence of water supply interruptions [20], the availability of water at home varied and depended on whether a household stored water or not. Some households may also purchase water to cope with problems encountered during a water interruption. Lack of available water at home is not only a barrier to latrine cleaning, but also may be a cause of diarrheal disease among slum-living children under five [20]. For instance, the 2017 WHO/UNICEF Joint Monitoring Program (JMP) report indicated that in 2015, 29.0% of the global population (2.1 billion people) lacked properly managed drinking water services and 61.0% (4.5 billion people) lacked adequate sanitation services [30], which compromises latrine cleaning practices. A study in Tanzania showed water scarcity was a barrier to a national sanitation campaign [45].

## Limitations of the study

One of the limitations of this study was that the unmatched case–control study design did not control confounders at the design stage, whereas a matched case–control design may provide better evidence due to the ability to control potential confounders at the design stage. The self-reported data may have bias by underestimating or overestimating the sanitation and hygiene status of the latrine [46], although most of the data was collected by data collectors using an observational checklist. A follow-up study designed with a mixed method of data (quantitative and qualitative) may help to capture more of the practical situation.

## Conclusion

We found that the barriers to cleaning a shared latrine were monthly household income, users feeling a lack of privacy during latrine use, lack of a locking latch on the shared latrine door,

inadequate latrine ventilation, a lack of regular monitoring of a shared latrine by a health extension worker and a lack of enough water at home for cleaning the latrine.

We recommend that the Addis Ababa Water and Sewerage Authority and other concerned bodies implement the Urban Sanitation Strategic Plan in a well-organized and integrated manner that gives special attention to needs of the shared latrine users in slums of Addis Ababa. Promoting the cleaning of shared latrines through designing strategies for increasing the household monthly income by means of the Urban Productive Safety Net program, increasing the privacy of the latrines by fixing the superstructure, providing latrines with doors and locking latches, adjusting the latrines to allow adequate ventilation, regular monitoring of the latrines by health extension workers and by making enough water available for regular latrine cleaning. Previous findings also have shown that sanitation issues affecting all community members can be addressed in meetings that ensure collective decision making and formulation of rules for sanitation facilities [47].

## Supporting information

**S1 Questionnaire. Survey tool in English language.**
(DOCX)

**S2 Questionnaire. Survey tool in Amharic (local) language.**
(DOCX)

**S1 Data.**
(DTA)

## Acknowledgments

We thank GAMBY College of Medical and Business Sciences for the support we received from the inception to the finalization of the study. We gratefully acknowledge District 05 in Lideta Sub-City Health Office for their cooperation during data collection. We also thank study participants, data collectors and supervisors for their kind assistance during the study. We also highly appreciate Lisa Penttila for language editing of this manuscript.

## Author Contributions

**Conceptualization:** Kidist Hailu, Zewdie Aderaw Alemu, Metadel Adane.

**Data curation:** Kidist Hailu, Metadel Adane.

**Formal analysis:** Kidist Hailu, Metadel Adane.

**Investigation:** Kidist Hailu, Zewdie Aderaw Alemu, Metadel Adane.

**Methodology:** Kidist Hailu, Zewdie Aderaw Alemu, Metadel Adane.

**Project administration:** Kidist Hailu, Metadel Adane.

**Resources:** Kidist Hailu, Zewdie Aderaw Alemu, Metadel Adane.

**Software:** Zewdie Aderaw Alemu, Metadel Adane.

**Supervision:** Zewdie Aderaw Alemu, Metadel Adane.

**Validation:** Metadel Adane.

**Visualization:** Metadel Adane.

**Writing – original draft:** Metadel Adane.

**Writing – review & editing:** Metadel Adane.

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
