## [Decision Letter · Decision Letter 0]

12 Oct 2020

PONE-D-20-23172

Barriers to Cleaning of Shared Latrines in Addis Ababa Slums: Unconditional Logistic Regression Analysis

PLOS ONE

Dear Dr. Adane (PhD),

Thank you for submitting your manuscript to PLOS ONE. After careful consideration, we feel that it has merit but does not fully meet PLOS ONE’s publication criteria as it currently stands. Therefore, we invite you to submit a revised version of the manuscript that addresses the points raised during the review process.

We look forward to receiving your revised manuscript.

Kind regards,

Hans-Uwe Dahms, Ph.D.

Academic Editor

PLOS ONE

Journal Requirements:

2. Please clarify the relationshop between the authors and the IRB. We typically expect at least one author to be affiliated with the institution which provides ethical oversight.

4. Please include additional information regarding the survey or questionnaire used in the study and ensure that you have provided sufficient details that others could replicate the analyses.

For instance, if you developed a questionnaire as part of this study and it is not under a copyright more restrictive than CC-BY, please include a copy, in both the original language and English, as Supporting Information.

5. Please amend either the title on the online submission form (via Edit Submission) or the title in the manuscript so that they are identical.

6. We note that Figure 1 in your submission contains map images which may be copyrighted.

We require you to either (a) present written permission from the copyright holder to publish these figure specifically under the CC BY 4.0 license, or (b) remove the figure from your submission:

b. If you are unable to obtain permission from the original copyright holder to publish these figure under the CC BY 4.0 license or if the copyright holder’s requirements are incompatible with the CC BY 4.0 license, please either i) remove the figure or ii) supply a replacement figure that complies with the CC BY 4.0 license. Please check copyright information on all replacement figures and update the figure caption with source information. If applicable, please specify in the figure caption text when a figure is similar but not identical to the original image and is therefore for illustrative purposes only.

Additional Editor Comments:

Pls. consider and REVISE your MS particularly following REVIEWER 1 who otherwise rejected your contribution!

This study uses an unmatched case-control design to assess barriers to cleaning of community latrines in Addis Ababa. Cases were those who did not clean latrines, and controls were those who did clean latrines based on self-report. Overall more information is required around how/why certain variables were selected for use in the logistic regression models (why demographic variables such as gender were not included) as well as details around recruitment methods. Finally, there is some confusion around the differences between data that were collected by observation compared to data collected in surveys, throughout the manuscript. Ideally these data should be reported an analyzed separately to reduce this confusion. My specific comments on each section of the manuscript are included below:

Background

In general, I find the Background section to be a bit too lengthy and I think that some content can be removed or shortened.

The final paragraph should include a succinct description of the study’s purpose (lines 126-127) which includes the study design and data used.

Line 70-71: “Shared facilities can reduce stress when proper maintenance and management systems are in place.” Please define what is meant by “stress” in this context.

Line 96-96: “Strina et al. found that people in latrine-owning 97 households in Salvador, Brazil behaved more hygienically than those without latrine.” Is this study looking at household latrines or shared latrines? This difference is very important to the underlying purpose of the manuscript, please be explicit.

Line 105-107: “Improving cleaning practices of shared latrines is a step toward achieving the United Nations’ 2030 goals for Sustainable Development, in line with achieving Target 6.2 of the Sustainable Development Goal (SDG) of universal access to sanitation as a key priority [29].” I think it may be useful to write out the exact wording of the Target and discuss how it does or does not apply to shared latrines.

Methods

Include a description around how study participants were recruited from the source population. How was the study introduced during the house-to-house visits, was there an IRB process?

How many individuals were considered non-respondents (line 173)?

Please describe what is meant by “regular monitoring by health extension workers”? Who is in charge of this monitoring and who determines which latrines are monitored?

In the paragraph beginning on line 225, please further explain how “validity” was assessed. Was it based on qualitative measurements? Provide more detail here.

In the description of the statistical analysis, explain how covariates were selected for the multivariable analysis. Why were certain questions included on the survey and in the model? Were there any survey questions that were not included in the final adjusted models?

The authors note that they assess multi-collinearity using standard errors, although this is not the correct method. The authors should calculate the variance inflation factor (VIF) and report those values.

Results

Line 316: explain why the cut-off income was $55.60, how was that value chosen?

Table 1: why is “divorced” the reference group in the logistic regression? The reference group is typically one of the more common groups, e.g., married or single. Same comment applies for the referent group select from the educational status and occupation variables.

Why is the “household size” variable entered as a binary variable rather than a continuous number of individuals per household measure?

Because you note that individuals in both the case and control group may be using the same latrines, I think what the survey is measuring is latrine perceptions rather than actual latrines data. For example, there were significant differences between cases and controls in reporting on privacy and whether the latrine had a door. If they are using the same latrines than this difference is based on their perceptions rather than actuality as there would be no difference there.

Do men and women use the same latrines? If not, analyses should be stratified by male/female respondent as they would be assessing a different set of latrines.

It is unclear throughout which data were collected in the surveys with participants and which data were collected using the observational checklist. This leads to a lot of confusion around whether you are talking about the latrine itself or individuals’ perceptions. These sets of data should have different samples sizes and should be reported and analyzed separately.

Discussion

Line 388 “This study used an unmatched case-control design and controlling of the confounders at the design stage was not possible.” Why was this not possible in this context? Also, why were no confounders (gender, age, etc) assessed during the analytic stage?

Finally, the manuscript included several grammatic mistakes throughout which should be addressed. For example:

- In Abstract: “barriers to keeping shared latrines cleaning”

- Line 138-139: “80% of Addis Ababa was slums”

- Line 371: “not regularly monitoring of the latrine”

Reviewers' comments:

Reviewer's Responses to Questions

**Comments to the Author**

1. Is the manuscript technically sound, and do the data support the conclusions?

Reviewer #1: No

Reviewer #2: Partly

2. Has the statistical analysis been performed appropriately and rigorously? 

Reviewer #1: No

Reviewer #2: Yes

3. Have the authors made all data underlying the findings in their manuscript fully available?

Reviewer #1: No

Reviewer #2: Yes

4. Is the manuscript presented in an intelligible fashion and written in standard English?

Reviewer #1: No

Reviewer #2: Yes

5. Review Comments to the Author

Reviewer #1: The authors investigated the barriers to cleaning shared latrines in Addis Ababa, Ethiopia. The authors conducted a rigorous household survey and attempt to explain their findings in the manuscript. I commend the authors for the substantial improvements to the manuscript. In particular, the manuscript is substantially improved in its engagement with the broader literature and the writing is improved. As with the previous manuscript, the study appears well designed and the data sound, but I still find the manuscript unsuitable for publication in its current form. Another round of substantial revisions are needed. The manuscript is very repetitive, there are too many details in the abstract, the writing unclear at certain points, there are numerous grammatical mistakes, the statistical analysis is incomplete and unclear, there is little discussion on the limitations, and the some of the recommendations are generic. Further, I would suggest you make use of the supplemental information to make the manuscript more concise.

Main Comments

1. There are too many details in the abstract. Line 30-37 is too much detail. Line 46-50 is a repetition of the previous sentence.

2. Line 35. Here and throughout you never mention what covariates you used in your adjusted analysis

3. Line 50-54: These recommendations are too generic. This manuscript offers an opportunity to make credible suggestions to an interested audience. Increasing household income is a noble aim but is not a feasible recommendation from this data.

4. The introduction is insufficiently focused and should not be a mini-review of the broader field.

5. Line 145: How were study participants selected? Was it every member of District 05 (line 152)? This is unclear to me.

6. Again on Line 168. Did you visit every household?

7. Lines 218-223: Is this the only data that you collected? Or are you only reporting the statistically significant data? It seems like you ran a lot of models, and to avoid accusations of p-hacking, you should adjust your analysis for multiple comparisons.

8. Line 253. This citation is not sufficient here. You need to explain what covariates you used in your adjusted models.

9. Line 316: You calculated odds ratios, not risk ratios. Please make sure the language you use accurately explains your results. As written, this is incorrect.

10. You calculated odds ratios, which are a measure of association, not causation. The discussion section needs to reflect this.

11. Line 334: All the associations you observed could be explain by poverty. If you did not adequately adjust for household wealth or income in your analysis, these finding may be spurious.

12. Line 388: Further discussion of the limitations of this study are needed. What potential sources of bias may have influenced your results?

13. The language in the conclusion needs to better reflect that you found associations, not causation

14. This manuscript may be more suited for a more specialized journal, such as the Journal of Water, Sanitation, and Hygiene for Development

Minor Comments

1. Line 25: As currently written, you are implying there currently is a comprehensive sanitation program in this setting? Is that true?

2. Line 42-46. Why do you use 1 significant figure for some values and 2 significant figures for others?

3. Line 65: All countries are developing. I would suggest you refer to these countries as low- and middle-income (LMIC)

4. Line 69: How common are shared latrines? Can you cite a specific number? (See Berendes et al. 2017 10.1021/acs.est.6b06019)

5. Line 70: Substitute “may” for “can”

6. Line 78: The rainy season where? Or are you referencing rainy seasons in general?

7. Line 84-85: You mentioned the number of users here, but don’t discuss this later in the manuscript. Do you have this data? This seems like an important gap missing from you manuscript

8. Line 86-87: You mention users working collectively here. Did you collect any data on how users work together to clean their systems?

9. Lines 95-103: This reads like a list of studies rather than a concise introduction

10. Line 107-109: repetitive

11. Line 113: Disease and infection are not the same.

12. Line 140-141: You define public latrines later, so at the point it is unclear whether public latrines and shared latrines are the same thing

13. Line 172-173. Here you talk about non-respondents, but later you say you had 0 non-responses (Line 276). Which is it?

14. Line 173-177: This level of detail can go in the supplemental

15. Line 180: I would suggest you work these definitions into the manuscript naturally and can include these full definitions in the supplemental

16. Line 203: People may be illegally occupying land, but the people themselves are not illegal. I would suggest revising this definition

17. Line 212: too much detail

18. Line 276: What do you mean you “employed” study participants? Did you pay them?

19. Line 277: As you had a different number of cases and control I find it very confusing that you report your results as number(percentage). At minimum you need to included the denominator to these values. Since the sample size is different, it is much easier to compare the percentages than the raw values.

20. Line 300: Why may have some households been visited more often than others by the health extension workers? I expected to see this in the discussion

21. Line 306: Why may some households have better water access than others? Did you include household wealth or income as a covariate? I would assume wealthier households had better water access

22. Line 316: Can you provide a citation for why choose to compare income above and below the mean? I would have expected this to be quartiles or quintiles.

23. Line 319: You say “less likely” but the AOR is greater than 1. This data should be reported saying something like “Households who reported feeling privacy in their latrine had X times the odds of reporting to clean their latrine in the previous week compared to …”

24. Line 348: But was privacy associated with household income? Perhaps only high-income households could afford to build a private latrine.

25. Line 371: I find this interesting. Who are the health extension workers? How often do they typically visit? Why did they only visit some households?

26. Line 380: What was the water availability? Did all participants get their water from the same source? Additional context is needed in the results to contextualize this finding

Reviewer #2: This study uses an unmatched case-control design to assess barriers to cleaning of community latrines in Addis Ababa. Cases were those who did not clean latrines, and controls were those who did clean latrines based on self-report. Overall more information is required around how/why certain variables were selected for use in the logistic regression models (why demographic variables such as gender were not included) as well as details around recruitment methods. Finally, there is some confusion around the differences between data that were collected by observation compared to data collected in surveys, throughout the manuscript. Ideally these data should be reported an analyzed separately to reduce this confusion. My specific comments on each section of the manuscript are included below:

Background

In general, I find the Background section to be a bit too lengthy and I think that some content can be removed or shortened.

The final paragraph should include a succinct description of the study’s purpose (lines 126-127) which includes the study design and data used.

Line 70-71: “Shared facilities can reduce stress when proper maintenance and management systems are in place.” Please define what is meant by “stress” in this context.

Line 96-96: “Strina et al. found that people in latrine-owning 97 households in Salvador, Brazil behaved more hygienically than those without latrine.” Is this study looking at household latrines or shared latrines? This difference is very important to the underlying purpose of the manuscript, please be explicit.

Line 105-107: “Improving cleaning practices of shared latrines is a step toward achieving the United Nations’ 2030 goals for Sustainable Development, in line with achieving Target 6.2 of the Sustainable Development Goal (SDG) of universal access to sanitation as a key priority [29].” I think it may be useful to write out the exact wording of the Target and discuss how it does or does not apply to shared latrines.

Methods

Include a description around how study participants were recruited from the source population. How was the study introduced during the house-to-house visits, was there an IRB process?

How many individuals were considered non-respondents (line 173)?

Please describe what is meant by “regular monitoring by health extension workers”? Who is in charge of this monitoring and who determines which latrines are monitored?

In the paragraph beginning on line 225, please further explain how “validity” was assessed. Was it based on qualitative measurements? Provide more detail here.

In the description of the statistical analysis, explain how covariates were selected for the multivariable analysis. Why were certain questions included on the survey and in the model? Were there any survey questions that were not included in the final adjusted models?

The authors note that they assess multi-collinearity using standard errors, although this is not the correct method. The authors should calculate the variance inflation factor (VIF) and report those values.

Results

Line 316: explain why the cut-off income was $55.60, how was that value chosen?

Table 1: why is “divorced” the reference group in the logistic regression? The reference group is typically one of the more common groups, e.g., married or single. Same comment applies for the referent group select from the educational status and occupation variables.

Why is the “household size” variable entered as a binary variable rather than a continuous number of individuals per household measure?

Because you note that individuals in both the case and control group may be using the same latrines, I think what the survey is measuring is latrine perceptions rather than actual latrines data. For example, there were significant differences between cases and controls in reporting on privacy and whether the latrine had a door. If they are using the same latrines than this difference is based on their perceptions rather than actuality as there would be no difference there.

Do men and women use the same latrines? If not, analyses should be stratified by male/female respondent as they would be assessing a different set of latrines.

It is unclear throughout which data were collected in the surveys with participants and which data were collected using the observational checklist. This leads to a lot of confusion around whether you are talking about the latrine itself or individuals’ perceptions. These sets of data should have different samples sizes and should be reported and analyzed separately.

Discussion

Line 388 “This study used an unmatched case-control design and controlling of the confounders at the design stage was not possible.” Why was this not possible in this context? Also, why were no confounders (gender, age, etc) assessed during the analytic stage?

Finally, the manuscript included several grammatic mistakes throughout which should be addressed. For example:

- In Abstract: “barriers to keeping shared latrines cleaning”

- Line 138-139: “80% of Addis Ababa was slums”

- Line 371: “not regularly monitoring of the latrine”

6. PLOS authors have the option to publish the peer review history of their article (what does this mean?). If published, this will include your full peer review and any attached files.

Reviewer #1: No

Reviewer #2: No

---

## [Author Response · Author response to Decision Letter 0]

29 Dec 2021

Line by line response to Reviewers for PONE-D-20-23172

Corresponding authors: Metadel Adane et al. 

Dear Professor Hans-Uwe Dahms,

Thank you for your letter dated October 12, 2020 with a decision of revision needed. We were pleased to know that our manuscript was considered potentially acceptable for publication in PLoS ONE, subject to adequate revision as requested by the reviewers. Based on the instructions provided in your letter, we uploaded the file of the rebuttal letter; the marked-up copy of the revised manuscript highlighting the changes made in the original submitted version and the clean copy of the revised manuscript. 

We addressed 67 questions/comments raised by the journal office staff, editor and two reviews. We are thankful for having so many questions. We have revised the manuscript by modifying the abstract, introduction, methods, results, discussion and other sections, based on the comments made by the reviewers and using the journal guidelines. Accordingly, we have marked in red color all the changes made during the revision process. Appended to this letter is our point-by-point response (rebuttal letter) to the comments made by the reviewers. 

We agree with almost all the comments/questions raised by the reviewers and provided justification for disagreeing with some of them. We would like to take this opportunity to express our thanks to the reviewers for their valuable comments and to thank you for allowing us to resubmit a revision of the manuscript.

I hope that the revised manuscript is accepted for publication in PLoS ONE. 

I am so sorry for being late in responding due to the COVID-19 crises. 

Sincerely yours,

Metadel Adane (PhD) 

Response to Journal office requirements 

Q1. Please ensure that your manuscript meets PLOS ONE's style requirements, including those for file naming. 

Answer: Thank you for this key comment: We formatted the manuscript with the PLoS ONE style. Please see the updated version. 

Q2. Please clarify the relationship between the authors and the IRB. We typically expect at least one author to be affiliated with the institution which provides ethical oversight.

Answer: Thank you. The Ethical Considerations section has been updated to clarify this point. One of the significant contributors to the paper, Zewdie Aderaw Alemu, is affiliated with the GAMBY Medical and Business College, whose Institutional Review Board provided ethical clearance for the study. This contributor has been named as a co-author. Please see the revised ethical statement letter and the co-author affiliation. 

Q3. We suggest you thoroughly copyedit your manuscript for language usage, spelling, and grammar. If you do not know anyone who can help you do this, you may wish to consider employing a professional scientific editing service. 

Response: We have received the support of a professional language editor whose name is Lisa Penttila from Canada. We hope that the language usage, spelling, and grammar is now acceptable. Please see the updated manuscript.

Q4. Please include additional information regarding the survey or questionnaire used in the study and ensure that you have provided sufficient details that others could replicate the analyses.

Response: Thank you for this important comment. We provided detailed information regarding the survey/questionnaire used in the study in the Method section of the page 9 from lines 194 and 195. This will help to ensure that others could replicate the analyses. We also included the questionnaire as supporting information SI and SII. Please see page 19. 

Q5. Please amend either the title on the online submission form (via Edit Submission) or the title in the manuscript so that they are identical. 

Response: We have made the amendment to ensure the title is consistent between the manuscript and online submission system. Thank you.

Q6. We note that Figure 1 in your submission contains map images which may be copyrighted.

Response: Thank you for your concern. Figure 1 has been deleted since the text contains the key geographic information. 

Response the academic editor 

Additional Editor Comments:

Q1. Pls. consider and REVISE your MS particularly following REVIEWER 1 who otherwise rejected your contribution!

Response: Thank you. We carefully addressed reviewer 1 and other reviewer comments.

Q2. This study uses an unmatched case-control design to assess barriers to cleaning of community latrines in Addis Ababa. Cases were those who did not clean latrines, and controls were those who did clean latrines based on self-report. Overall more information is required around how/why certain variables were selected for use in the logistic regression models (why demographic variables such as gender were not included) as well as details around recruitment methods. Finally, there is some confusion around the differences between data that were collected by observation compared to data collected in surveys, throughout the manuscript. Ideally these data should be reported an analyzed separately to reduce this confusion. My specific comments on each section of the manuscript are included below:

Response: Dear Prof Hans, thank you for your comment. Gender was excluded from the multivariable analysis due to the fact that the p-value from the bivariate analysis was greater than 2.5. Variables were selected to the multivariable analysis only if the p-value from the bivariate analysis was less than 0.25. 

We also briefly pointed out that data was measured by observation and interviewing in the Method section on page 11 from lines 249 to 256-. However, after proper categorization of variables, both types of variables (those measured by observation and by interview) were considered in the same data reporting and analyzed in the same model of unconditional logistic regression, which is also the most common way of data management during logistic regression. Please see below for several similar studies that I handled in this way.

Similar papers link about 

• https://doi.org/10.1371/journal.pone.0182783

• https://doi.org/10.1371/journal.pone.0181516

• DOI: 10.1007/s10900-017-0437-1

Background

Q3. In general, I find the Background section to be a bit too lengthy and I think that some content can be removed or shortened.

Response: Thank you for this key feedback. We revised the background section as suggested. Please see the Background of the revised version. We reduced the length by about a third. 

Q4. The final paragraph should include a succinct description of the study’s purpose (lines 126-127) which includes the study design and data used.

Response: We included the purpose of the study, study design and data used. Please see the last paragraph of the Background section. 

Q5. Line 70-71: “Shared facilities can reduce stress when proper maintenance and management systems are in place.” Please define what is meant by “stress” in this context.

Response: To clarify this sentence, we replaced the term “stress” with another positive term. See page --- lines -----

Q6. Line 96-96: “Strina et al. found that people in latrine-owning 97 households in Salvador, Brazil behaved more hygienically than those without latrine.” Is this study looking at household latrines or shared latrines? This difference is very important to the underlying purpose of the manuscript, please be explicit.

Response: Thank you for this key comment. Since the study of Strina et al. was about household latrines and our study is about shared latrines, we deleted the idea of Strina et al. 

Q7. Line 105-107: “Improving cleaning practices of shared latrines is a step toward achieving the United Nations’ 2030 goals for Sustainable Development, in line with achieving Target 6.2 of the Sustainable Development Goal (SDG) of universal access to sanitation as a key priority [29].” I think it may be useful to write out the exact wording of the Target and discuss how it does or does not apply to shared latrines.

Response: Thank you for this valuable comment. In order to make the Background section shorter as suggested above, ideas about SDGs were deleted.

Methods

Q8. Include a description around how study participants were recruited from the source population. How was the study introduced during the house-to-house visits, was there an IRB process?

Response: Thank you for this important comment. We provided detailed information about the study participant recruitment, house-to-house visits and about ethical consideration. Please see the Method section on page 9 about study participant recruitment, page 10 about home visits and page 11 about IRB process. 

Q9. How many individuals were considered non-respondents (line 173)?

Response: Unfortunately in this study, there were no non-respondents, which might be due to the random selection of cases and controls based on the self-report of latrine cleaning condition (cleaned or not cleaned) until the sample size was achieved. We provided detailed information about this in the description of the sampling procedure on page 9. 

Q10. Please describe what is meant by “regular monitoring by health extension workers”? Who is in charge of this monitoring and who determines which latrines are monitored?

Response: Regular monitoring of latrines by health extension workers is a part of Ethiopia’s community-based Health Extension Program. Each district health bureau assigns a health extension worker to monitor the execution of Health Extension Package programs such as solid waste management, latrine cleaning, water handling practice and so on. The district health bureau tracks the weekly and monthly reports of each health extension worker, while each district health bureau is in turn monitored by the sub-city health office. 

Q11. In the paragraph beginning on line 225, please further explain how “validity” was assessed. Was it based on qualitative measurements? Provide more detail here.

Response: Thank you for bringing attention to this key issue. To ensure the quality of the data, we also pre-tested the questionnaire among 10 cases and 20 control households (10% of the sample size) in one non-selected area of District 4 in Lideta Sub-City to evaluate its face and content validity. During the pre-test, face validity was checked to verify that the questions measured what they were intended to measure. Content validity was also checked by three experts re (environmental health professionals) who ensured that the survey contained questions that covered all aspects of the construct being measured. Data collection began after we approved the face and content validity of the survey tool.

Q12. In the description of the statistical analysis, explain how covariates were selected for the multivariable analysis. Why were certain questions included on the survey and in the model? Were there any survey questions that were not included in the final adjusted models?

Response: For variable selection to the final model, variables (covariates) with a p-value less than 0.25 from bivariate analysis were included to the multivariable analysis. Some variables such as sex and age were not included in the final model due to the fact that the p-value from the bivariate analysis were greater than or equal to 0.25. Please see the detailed information about data analysis on page 11. 

Q13. The authors note that they assess multi-collinearity using standard errors, although this is not the correct method. The authors should calculate the variance inflation factor (VIF) and report those values.

Response: The presence of multi-collinearity among independent variables was also checked using variance inflation factor (VIF) and we found a maximum VIF of 2.0, which indicated there was no multi-collinearity between independent variables.

Results

Q14. Line 316: explain why the cut-off income was $55.60, how was that value chosen?

Response: The cut-off income $55.6 was determined based on the mean value of the income. 

Table 1: why is “divorced” the reference group in the logistic regression? The reference group is typically one of the more common groups, e.g., married or single. Same comment applies for the referent group select from the educational status and occupation variables.

Response: We selected the reference category depending on the direction of the association found during the analysis. When we selected the most frequent common group, it was found to be a preventive factor, which was odds ratio and confidence interval below 1. However, since we were studying associated factors, direction of association should be towards positive rather than explaining the findings of the preventive factors that has a negative direction of association. 

Q15. Why is the “household size” variable entered as a binary variable rather than a continuous number of individuals per household measure?

Response: During the binary logistic regression model analysis, continuous variables must be changed to categorical variables. Continuous variables such as household size were entered to the model if the model of data analysis used linear regression. But since our model used binary logistic regression, variables entered to the model had to have a certain category with reference group. Following this rule for logistic regression, continuous variables were changed to categorical variables. 

Q16. Because you note that individuals in both the case and control group may be using the same latrines, I think what the survey is measuring is latrine perceptions rather than actual latrines data. For example, there were significant differences between cases and controls in reporting on privacy and whether the latrine had a door. If they are using the same latrines than this difference is based on their perceptions rather than actuality as there would be no difference there.

Response: Thank you for this concern. In this study, that cases and control should not use the same latrine was one of the criteria used during selection of case and control. Wording has been revised to make this point clear.

Q17. Do men and women use the same latrines? If not, analyses should be stratified by male/female respondent as they would be assessing a different set of latrines.

Response: Yes, men and women used the same latrine if they lived in the same household or if their households shared the same latrine. In slums of Addis Ababa, shared latrines are not classified for men or women, although this division exists in hotels, cafes and restaurants. 

Q18. It is unclear throughout which data were collected in the surveys with participants and which data were collected using the observational checklist. This leads to a lot of confusion around whether you are talking about the latrine itself or individuals’ perceptions. These sets of data should have different samples sizes and should be reported and analyzed separately.

Response: We provided detail information about variables collected by observation and interviewing. Please see page 9. Thank you for this valuable concern. 

Discussion

Q19. Line 388 “This study used an unmatched case-control design and controlling of the confounders at the design stage was not possible.” Why was this not possible in this context? Also, why were no confounders (gender, age, etc) assessed during the analytic stage?

Response: Although controlling of the confounders at the design stage would be performed for a matched case-control study, our study used an unmatched case-control design, so confounders were controlled during data analysis, but not at the design stage. This is a principle of the difference between matched and unmatched case-control designs. However, in our study, confounders such as gender and age were assessed during the analysis stage. Thus, one of the limitations of this study was its unmatched case-control study design, which did not help to control confounders at the design stage; whereas using matched case-control design may provide better evidence due to the fact that potential confounders can be controlled at the design stage during matching of cases and controls using certain expected confounder variables. 

Q20. Finally, the manuscript included several grammatic mistakes throughout which should be addressed. For example:

- In Abstract: “barriers to keeping shared latrines cleaning”

- Line 138-139: “80% of Addis Ababa was slums”

- Line 371: “not regularly monitoring of the latrine”

Response: We regret these errors and have made revisions as noted. 

The word ”keeping” was deleted in the Abstract.

“80% of Addis Ababa was slums” was changed to “four-fifths (80%) of Addis Ababa was classified as slums”. 

The noted phrase “not regularly monitoring of the latrine” was changed to “not regularly monitoring the latrine”.

Line by line response to reviewers 

Reviewer #1: 

Q1. The authors investigated the barriers to cleaning shared latrines in Addis Ababa, Ethiopia. The authors conducted a rigorous household survey and attempt to explain their findings in the manuscript. I commend the authors for the substantial improvements to the manuscript. In particular, the manuscript is substantially improved in its engagement with the broader literature and the writing is improved. As with the previous manuscript, the study appears well designed and the data sound, but I still find the manuscript unsuitable for publication in its current form. Another round of substantial revisions are needed. The manuscript is very repetitive, there are too many details in the abstract, the writing unclear at certain points, there are numerous grammatical mistakes, the statistical analysis is incomplete and unclear, there is little discussion on the limitations, and the some of the recommendations are generic. Further, I would suggest you make use of the supplemental information to make the manuscript more concise.

Response: We really appreciate your scientific input. We updated the Abstract, minimized the repetition in the Background section, corrected grammatical errors and also we made revisions to the Limitations section. Please see all sections in the revised manuscript. We also strengthened the specificity of the conclusion based on the findings. 

Main Comments

Q2. There are too many details in the abstract. Line 30-37 is too much detail. Line 46-50 is a repetition of the previous sentence.

Response: We minimize details in lines 30-37 and we also deleted line 46-50 to avoid repetition. Thank you. 

Q3. Line 35. Here and throughout you never mention what covariates you used in your adjusted analysis

Response: Sorry for the confusion. In the data analysis, we briefly mentioned the selection of covariates for the adjusted analysis. Please see the Method section on page 11. 

Q4. Line 50-54: These recommendations are too generic. This manuscript offers an opportunity to make credible suggestions to an interested audience. Increasing household income is a noble aim but is not a feasible recommendation from this data.

Response: We revised the recommendations. We deleted increasing household income as a recommendation in the abstract. Please see the revised version of the abstract. 

Q5. The introduction is insufficiently focused and should not be a mini-review of the broader field.

Response: We accept your comment. We revised the introduction to make it more focused. Please see the whole Introduction section. Thank you. 

Q6. Line 145: How were study participants selected? Was it every member of District 05 (line 152)? This is unclear to me.

Response: Thank you for this key comment. We provided detailed information about how the study participants selected. By house-to-house visits, cases were randomly selected from those who reported they had not cleaned the shared latrine at least once during the week prior to data collection and controls were selected from shared latrine users who reported they had cleaned the latrine at least once during the week prior to data collection. 

Study participants who were not available during the survey were revisited once on the same day or the next day. If not available again, a third visit was made and if not available again, the study participant would be considered a non-respondent; however, in this study there was no non-respondents, which might be due to the random selection of cases and controls based on the self-report of latrine cleaning condition (cleaned or not cleaned) before the sample size was achieved. See the sampling procedure on page 7 and 8. 

 Q7. Again on Line 168. Did you visit every household?

Response: Only households that used shared latrines were visited. Households that used private latrines were not part of the study and not visited. Households were visited during identification of cases and controls. 

Q8. Lines 218-223: Is this the only data that you collected? Or are you only reporting the statistically significant data? It seems like you ran a lot of models, and to avoid accusations of p-hacking, you should adjust your analysis for multiple comparisons.

Response: We listed variables for collected data in the Method section. However, during discussion, we only discussed for those significant variables. We ran only one model and adjusted analysis performed to control confounders. Please see the data analysis section on page 11.

Q9. Line 253. This citation is not sufficient here. You need to explain what covariates you used in your adjusted models.

Response: We fixed a cut-off point to include covariates for the adjusted model. Listing those variables in the data analysis would add unnecessary repetition of ideas, we believe. However, for variable selection to the final model, variables with a p-value less than 0.25 from bivariate analysis were included to the multivariable analysis.

Q10. Line 316: You calculated odds ratios, not risk ratios. Please make sure the language you use accurately explains your results. As written, this is incorrect.

Response: Thank you. We deleted the term “risk” throughout the paper. 

Q11. You calculated odds ratios, which are a measure of association, not causation. The discussion section needs to reflect this.

Response: Thank you. We updated the discussion accordingly. 

Q12. Line 334: All the associations you observed could be explain by poverty. If you did not adequately adjust for household wealth or income in your analysis, these finding may be spurious.

Response: We adjusted for income in the final model and found it is one of the factors for not cleaning the shared larine. We believe that our analysis is correct. We appreciate your comment. 

Q13. Line 388: Further discussion of the limitations of this study are needed. What potential sources of bias may have influenced your results?

Response: We improved the Limitations section as suggested. The self-reported data such as household income, may have bias by underestimate or overestimate. Furthermore, the sanitation and hygiene status of the latrine may be reported wrongly, although most of the data was collected using on-the-spot observational checklist. During observation, there may also have been bias of the data collector. For instance, latrines that were clean during data collection may have been classified as unclean. Please see the limitation on page 18. 

Q14. The language in the conclusion needs to better reflect that you found associations, not causation

Response: Thank you. We improved the conclusion with this in mind. 

Q15. This manuscript may be more suited for a more specialized journal, such as the Journal of Water, Sanitation, and Hygiene for Development

Response: Yes, this is reasonable comment. But we found that PLoS ONE has a high impact factor, whereas Journal of Water, Sanitation, and Hygiene for Development has a lower impact factor. We are interested in publishing in a high impact factor journal. 

Minor Comments: 

Q16. Line 25: As currently written, you are implying there currently is a comprehensive sanitation program in this setting? Is that true?

Response: No. We updated the sentence. Please see Abstract again. 

Q17. Line 42-46. Why do you use 1 significant figure for some values and 2 significant figures for others?

Response: Sorry for the confusion. We used two significant figures (two digits). Thank you. See the Abstract Result section. 

Q18. Line 65: All countries are developing. I would suggest you refer to these countries as low- and middle-income (LMIC)

Response: Thank you for this key comment. We updated as suggested. Please see the first line of the introduction. 

Q19. Line 69: How common are shared latrines? Can you cite a specific number? (See Berendes et al. 2017 10.1021/acs.est.6b06019)

Response: The term common has been deleted to avoid confusion. 

Q20. Line 70: Substitute “may” for “can”

Response: Done 

Q21. Line 78: The rainy season where? Or are you referencing rainy seasons in general?

Response: Yes, we are referencing rainy seasons in general. Thank you. 

Q22. Line 84-85: You mentioned the number of users here, but don’t discuss this later in the manuscript. Do you have this data? This seems like an important gap missing from you manuscript

Response: Thank you, we deleted it. 

Q23. Line 86-87: You mention users working collectively here. Did you collect any data on how users work together to clean their systems?

Response: To avoid confusion, the term “collectively” is changed to “together”. 9. Lines 95-103: Q24. This reads like a list of studies rather than a concise introduction

Response: Very nice feedback. We deleted this material. Please see the revised version. 

Q25. 10. Line 107-109: repetitive

Response: We agreed and deleted it. 

Q26. Line 113: Disease and infection are not the same.

Response: Yes, we deleted the term “infection” and made the information about “disease”. 

Q27. Line 140-141: You define public latrines later, so at the point it is unclear whether public latrines and shared latrines are the same thing

Response: Thank you. We revised the sentence. This study was conducted among shared latrine users. A public latrine is one type of shared latrine. Shared latrine means that a latrine is shared by two or more households. 

Q28. Line 172-173. Here you talk about non-respondents, but later you say you had 0 non-responses (Line 276). Which is it?

Response: Study participants who were not available during the survey were revisited once on the same day or the next day. If not available again, a third visit was made, and if not available again, the study participant was considered a non-respondent; however, in this study there was no non-respondent, which might be due to the random selection of cases and controls based on the self-report of latrine cleaning condition (cleaned or not cleaned) until the sample size achieved. 

Q29. Line 173-177: This level of detail can go in the supplemental

Response: We condensed the ideas of Line 173 to 177. Thank you. 

Q30. Line 180: I would suggest you work these definitions into the manuscript naturally and can include these full definitions in the supplemental

Response: Including the definitions in the supplemental may create confusion for readers once published. We feel that since the operational definitions are key for measurement, it is useful to include them in the Method section of the manuscript. 

Q31. Line 203: People may be illegally occupying land, but the people themselves are not illegal. I would suggest revising this definition

Response: We re-defined as “slums are areas dominated by informal settlements that are characterized by one or more of the five characteristics of overcrowding, poor sanitation, insecure land tenure, lack of access to water supply, poor housing quality and other infrastructure. 

Q32. Line 212: too much detail

Response: We tried to minimize it. Please see the revised version on page 

Q33. Line 276: What do you mean you “employed” study participants? Did you pay them?

Response: The term employed was deleted to avoid confusion. We mean that the study included 300 study participants. 

Q34. Line 277: As you had a different number of cases and control I find it very confusing that you report your results as number (percentage). At minimum you need to include the denominator to these values. Since the sample size is different, it is much easier to compare the percentages than the raw values.

Response: Since the sample size for cases were 100 and the denominator was 100, when the percentage was calculated for any number out of 100%, it would be similar. For example 30 becomes 30% and 40 become 40%. However, the percentage and number for controls are different since the denominator was 200. 

Q35. Line 300: Why may have some households been visited more often than others by the health extension workers? I expected to see this in the discussion

Response: We updated the Discussion section. Please see page 15 to 18. 

Q36. Line 306: Why may some households have better water access than others? Did you include household wealth or income as a covariate? I would assume wealthier households had better water access

Response: Thank you for this valuable comment. Access to water might depend on household income variation, as you mentioned. We considered income as one of the covariates and adjusted for analysis. We also found that low income is one of the factors for not cleaning the latrine. 

Q37. Line 316: Can you provide a citation for why choose to compare income above and below the mean? I would have expected this to be quartiles or quintiles.

Response: The study was conducted among the poor community in urban areas and considered only income and did not consider other assets. A wealth index that classifies in quartiles or quintiles should consider not only income but also other kind and in-kind assets. Therefore, we used only mean of the income to categorize variation. 

Q38. Line 319: You say “less likely” but the AOR is greater than 1. This data should be reported saying something like “Households who reported feeling privacy in their latrine had X times the odds of reporting to clean their latrine in the previous week compared to …”

Response: This was a major problem. We did the revision accordingly. We thank you so much. Please see the updated version on page 14. 

Q39. Line 348: But was privacy associated with household income? Perhaps only high-income households could afford to build a private latrine.

Response: Yes, privacy may be associated with household income, but in our case, we did not study the association between privacy and household income. However, in normal circumstances, they have a have direct relationship since high-income households could afford to build a private latrine.

Q40. Line 371: I find this interesting. Who are the health extension workers? How often do they typically visit? Why did they only visit some households?

Response: Health extension workers are health professionals assigned at kebele (administrative unit roughly equal to a neighborhood) level for improvement of the health system. This is a new innovation by the Ethiopia Ministry of Health where over 30,000 health extension workers are deployed throughout Ethiopia. The Health Extension Program is one the main health packages in Ethiopia. See the details about the Health Extension Program on page 17. 

Q41. Line 380: What was the water availability? Did all participants get their water from the same source? Additional context is needed in the results to contextualize this finding

Response: “This study also indicates that lack of water availability is one of the barriers to cleaning shared latrines. In our study setting, the source of the water is the same government supply for all. However, due to the presence of water supply interruptions, the availability of water at home varied and depended on whether a household stored water or not. Some households may also purchase water to cope with problems encountered during an interruption. Please see the Discussion section on page 17. 

Reviewer #2: 

Q1. Pls. consider and REVISE your MS particularly following REVIEWER 1 who otherwise rejected your contribution!

Response: Thank you. We carefully addressed reviewer 1 and other reviewer comments.

Q2. This study uses an unmatched case-control design to assess barriers to cleaning of community latrines in Addis Ababa. Cases were those who did not clean latrines, and controls were those who did clean latrines based on self-report. Overall more information is required around how/why certain variables were selected for use in the logistic regression models (why demographic variables such as gender were not included) as well as details around recruitment methods. Finally, there is some confusion around the differences between data that were collected by observation compared to data collected in surveys, throughout the manuscript. Ideally these data should be reported an analyzed separately to reduce this confusion. My specific comments on each section of the manuscript are included below:

Response: Dear Prof Hans, thank you for your comment. Gender was excluded from the multivariable analysis due to the fact that the p-value from the bivariate analysis was greater than 2.5. Variables were selected to the multivariable analysis only if the p-value from the bivariate analysis was less than 0.25. 

We also briefly pointed out that data was measured by observation and interviewing in the Method section on page 11 from lines 249 to 256-. However, after proper categorization of variables, both types of variables (those measured by observation and by interview) were considered in the same data reporting and analyzed in the same model of unconditional logistic regression, which is also the most common way of data management during logistic regression. Please see below for several similar studies that I handled in this way.

Similar papers link about 

• https://doi.org/10.1371/journal.pone.0182783

• https://doi.org/10.1371/journal.pone.0181516

• DOI: 10.1007/s10900-017-0437-1

Background

Q3. In general, I find the Background section to be a bit too lengthy and I think that some content can be removed or shortened.

Response: Thank you for this key feedback. We revised the background section as suggested. Please see the Background of the revised version. We reduced the length by about a third. 

Q4. The final paragraph should include a succinct description of the study’s purpose (lines 126-127) which includes the study design and data used.

Response: We included the purpose of the study, study design and data used. Please see the last paragraph of the Background section. 

Q5. Line 70-71: “Shared facilities can reduce stress when proper maintenance and management systems are in place.” Please define what is meant by “stress” in this context.

Response: To clarify this sentence, we replaced the term “stress” with another positive term. 

Q6. Line 96-96: “Strina et al. found that people in latrine-owning 97 households in Salvador, Brazil behaved more hygienically than those without latrine.” Is this study looking at household latrines or shared latrines? This difference is very important to the underlying purpose of the manuscript, please be explicit.

Response: Thank you for this key comment. Since the study of Strina et al. was about household latrines and our study is about shared latrines, we deleted the idea of Strina et al. 

Q7. Line 105-107: “Improving cleaning practices of shared latrines is a step toward achieving the United Nations’ 2030 goals for Sustainable Development, in line with achieving Target 6.2 of the Sustainable Development Goal (SDG) of universal access to sanitation as a key priority [29].” I think it may be useful to write out the exact wording of the Target and discuss how it does or does not apply to shared latrines.

Response: Thank you for this valuable comment. In order to make the Background section shorter as suggested above, ideas about SDGs were deleted.

Methods

Q8. Include a description around how study participants were recruited from the source population. How was the study introduced during the house-to-house visits, was there an IRB process?

Response: Thank you for this important comment. We provided detailed information about the study participant recruitment, house-to-house visits and about ethical consideration. Please see the Method section on page 9 about study participant recruitment, page 10 about home visits and page 11 about IRB process. 

Q9. How many individuals were considered non-respondents (line 173)?

Response: Unfortunately in this study, there were no non-respondents, which might be due to the random selection of cases and controls based on the self-report of latrine cleaning condition (cleaned or not cleaned) until the sample size was achieved. We provided detailed information about this in the description of the sampling procedure on page 9. 

Q10. Please describe what is meant by “regular monitoring by health extension workers”? Who is in charge of this monitoring and who determines which latrines are monitored?

Response: Regular monitoring of latrines by health extension workers is a part of Ethiopia’s community-based Health Extension Program. Each district health bureau assigns a health extension worker to monitor the execution of Health Extension Package programs such as solid waste management, latrine cleaning, water handling practice and so on. The district health bureau tracks the weekly and monthly reports of each health extension worker, while each district health bureau is in turn monitored by the sub-city health office. 

Q11. In the paragraph beginning on line 225, please further explain how “validity” was assessed. Was it based on qualitative measurements? Provide more detail here.

Response: Thank you for bringing attention to this key issue. To ensure the quality of the data, we also pre-tested the questionnaire among 10 cases and 20 control households (10% of the sample size) in one non-selected area of District 4 in Lideta Sub-City to evaluate its face and content validity. During the pre-test, face validity was checked to verify that the questions measured what they were intended to measure. Content validity was also checked by three experts re (environmental health professionals) who ensured that the survey contained questions that covered all aspects of the construct being measured. Data collection began after we approved the face and content validity of the survey tool.

Q12. In the description of the statistical analysis, explain how covariates were selected for the multivariable analysis. Why were certain questions included on the survey and in the model? Were there any survey questions that were not included in the final adjusted models?

Response: For variable selection to the final model, variables (covariates) with a p-value less than 0.25 from bivariate analysis were included to the multivariable analysis. Some variables such as sex and age were not included in the final model due to the fact that the p-value from the bivariate analysis were greater than or equal to 0.25. Please see the detailed information about data analysis on page 11. 

Q13. The authors note that they assess multi-collinearity using standard errors, although this is not the correct method. The authors should calculate the variance inflation factor (VIF) and report those values.

Response: The presence of multi-collinearity among independent variables was also checked using variance inflation factor (VIF) and we found a maximum VIF of 2.0, which indicated there was no multi-collinearity between independent variables.

Results

Q14. Line 316: explain why the cut-off income was $55.60, how was that value chosen?

Response: The cut-off income $55.6 was determined based on the mean value of the income. 

Table 1: why is “divorced” the reference group in the logistic regression? The reference group is typically one of the more common groups, e.g., married or single. Same comment applies for the referent group select from the educational status and occupation variables.

Response: We selected the reference category depending on the direction of the association found during the analysis. When we selected the most frequent common group, it was found to be a preventive factor, which was odds ratio and confidence interval below 1. However, since we were studying associated factors, direction of association should be towards positive rather than explaining the findings of the preventive factors that has a negative direction of association. 

Q15. Why is the “household size” variable entered as a binary variable rather than a continuous number of individuals per household measure?

Response: During the binary logistic regression model analysis, continuous variables must be changed to categorical variables. Continuous variables such as household size were entered to the model if the model of data analysis used linear regression. But since our model used binary logistic regression, variables entered to the model had to have a certain category with reference group. Following this rule for logistic regression, continuous variables were changed to categorical variables. 

Q16. Because you note that individuals in both the case and control group may be using the same latrines, I think what the survey is measuring is latrine perceptions rather than actual latrines data. For example, there were significant differences between cases and controls in reporting on privacy and whether the latrine had a door. If they are using the same latrines than this difference is based on their perceptions rather than actuality as there would be no difference there.

Response: Thank you for this concern. In this study, that cases and control should not use the same latrine was one of the criteria used during selection of case and control. Wording has been revised to make this point clear.

Q17. Do men and women use the same latrines? If not, analyses should be stratified by male/female respondent as they would be assessing a different set of latrines.

Response: Yes, men and women used the same latrine if they lived in the same household or if their households shared the same latrine. In slums of Addis Ababa, shared latrines are not classified for men or women, although this division exists in hotels, cafes and restaurants. 

Q18. It is unclear throughout which data were collected in the surveys with participants and which data were collected using the observational checklist. This leads to a lot of confusion around whether you are talking about the latrine itself or individuals’ perceptions. These sets of data should have different samples sizes and should be reported and analyzed separately.

Response: We provided detail information about variables collected by observation and interviewing. Please see page 9. Thank you for this valuable concern. 

Discussion

Q19. Line 388 “This study used an unmatched case-control design and controlling of the confounders at the design stage was not possible.” Why was this not possible in this context? Also, why were no confounders (gender, age, etc) assessed during the analytic stage?

Response: Although controlling of the confounders at the design stage would be performed for a matched case-control study, our study used an unmatched case-control design, so confounders were controlled during data analysis, but not at the design stage. This is a principle of the difference between matched and unmatched case-control designs. However, in our study, confounders such as gender and age were assessed during the analysis stage. Thus, one of the limitations of this study was its unmatched case-control study design, which did not help to control confounders at the design stage; whereas using matched case-control design may provide better evidence due to the fact that potential confounders can be controlled at the design stage during matching of cases and controls using certain expected confounder variables. 

Q20. Finally, the manuscript included several grammatic mistakes throughout which should be addressed. For example:

- In Abstract: “barriers to keeping shared latrines cleaning”

- Line 138-139: “80% of Addis Ababa was slums”

- Line 371: “not regularly monitoring of the latrine”

Response: We regret these errors and have made revisions as noted. 

The word ”keeping” was deleted in the Abstract.

“80% of Addis Ababa was slums” was changed to “four-fifths (80%) of Addis Ababa was classified as slums”. 

The noted phrase “not regularly monitoring of the latrine” was changed to “not regularly monitoring the latrine”.

---

## [Editor Report · Decision Letter 1]

19 Jan 2022

Barriers to Cleaning of Shared Latrines in Slums of Addis Ababa, Ethiopia

PONE-D-20-23172R1

Dear Dr. Adane,

We’re pleased to inform you that your manuscript has been judged scientifically suitable for publication and will be formally accepted for publication once it meets all outstanding technical requirements.

Kind regards,

Hans-Uwe Dahms, Ph.D.

Academic Editor

PLOS ONE

Additional Editor Comments (optional):

On the 12th October 12, 2020 I decided on major revision of this contribution.

The authors then addressed 67 questions/comments raised by the journal office staff, myself and two reviews.

The MS was now revised by modifying the abstract, introduction, methods, results, discussion and other sections,

based on the comments made by the reviewers and using the journal guidelines. The authors agreed with most of the

comments/questions raised by the reviewers and provided justification for disagreeing with some of them. The additional

author Zewdie Aderaw Alemu is accepted considering his contribution during the research work and during the revision

of the manuscript.END
---

## [Editor Report · Acceptance letter]

28 Feb 2022

PONE-D-20-23172R1 

Barriers to cleaning of shared latrines in slums of Addis Ababa, Ethiopia 

Dear Dr. Adane:

I'm pleased to inform you that your manuscript has been deemed suitable for publication in PLOS ONE. Congratulations! Your manuscript is now with our production department. 

Kind regards, 

on behalf of

Dr. Hans-Uwe Dahms 

Academic Editor

PLOS ONE